# Non-Gaussian noise spectroscopy with a superconducting qubit sensor

Youngkyu Sung [1,2], Félix Beaudoin [3,6], Leigh M. Norris[3], Fei Yan[1], David K. Kim[4], Jack Y. Qiu[1,2], Uwe von Lüpke[1], Jonilyn L. Yoder[4], Terry P. Orlando[1,2], Simon Gustavsson[1], Lorenza Viola [3] & William D. Oliver[1,2,4,5]

Accurate characterization of the noise influencing a quantum system of interest has far-reaching implications across quantum science, ranging from microscopic modeling of decoherence dynamics to noise-optimized quantum control. While the assumption that noise obeys Gaussian statistics is commonly employed, noise is generically non-Gaussian in nature. In particular, the Gaussian approximation breaks down whenever a qubit is strongly coupled to discrete noise sources or has a non-linear response to the environmental degrees of freedom. Thus, in order to both scrutinize the applicability of the Gaussian assumption and capture distinctive non-Gaussian signatures, a tool for characterizing non-Gaussian noise is essential. Here, we experimentally validate a quantum control protocol which, in addition to the spectrum, reconstructs the leading higher-order spectrum of engineered non-Gaussian dephasing noise using a superconducting qubit as a sensor. This first experimental demonstration of non-Gaussian noise spectroscopy represents a major step toward demonstrating a complete spectral estimation toolbox for quantum devices.

[1] Research Laboratory of Electronics, Massachusetts Institute of Technology, Cambridge, MA 02139, USA. [2] Department of Electrical Engineering and Computer Science, Massachusetts Institute of Technology, Cambridge, MA 02139, USA. [3] Department of Physics and Astronomy, Dartmouth College, Hanover, NH 03755, USA. [4] MIT Lincoln Laboratory, 244 Wood Street, Lexington, MA 02421, USA. [5] Department of Physics, Massachusetts Institute of Technology, Cambridge, MA 02139, USA. [6] Present address: NanoAcademic Technologies, 666 rue Sherbrooke Ouest, Suite 802, Montreal, Quebec H3A 1E7, Canada. Correspondence and requests for materials should be addressed to L.V. (email: lorenza.viola@dartmouth.edu) or to W.D.O. (email: william.oliver@mit.edu)

For any dynamical system that evolves in the presence of unwanted disturbances, precise knowledge of the noise spectral features is fundamental for quantitative understanding and prediction of the dynamics under realistic conditions. As a result, spectral estimation techniques have a long tradition and play a central role in classical statistical signal processing[1]. For quantum systems, the importance of precisely characterizing noise effects is further heightened by the challenge of harnessing the practical potential that quantum science and technology applications promise. Such detailed knowledge is key to develop noise-optimized strategies for enhancing quantum coherence and boosting control fidelity in near-term intermediate-scale quantum information processors[2], as well as for overcoming noise effects in quantum metrology[3,4]. Ultimately, probing the extent and decay of noise correlations will prove crucial in determining the viability of large-scale fault-tolerant quantum computation[5].

Thanks to their exquisite sensitivity to the surrounding environment, qubits driven by external control fields are naturally suited as "spectrometers", or sensors, of their own noise[6,7]. Quantum noise spectroscopy (QNS) leverages the fact that open-loop control modulation is akin to shaping the filter function (FF) that determines the sensor's response in frequency space[8-12] and, in its simplest form, aims to characterize the spectral properties of environmental noise as sensed by a single qubit sensor. By now, QNS protocols employing both pulsed and continuous control modalities have been explored, and experimental implementations have been reported across a wide variety of qubit platforms —including nuclear spins[13], superconducting quantum circuits[14-17], semiconductor quantum dots[18-21], diamond nitrogen vacancy centers[22,23], and trapped ions[24]. Notably, knowledge of the underlying noise spectrum has already enabled unprecedented coherence times to be achieved via tailored error suppression[25].

While the above advances clearly point to the growing significance of spectral estimation in the quantum setting, they all rely on the assumption that the target noise process is Gaussian— that is, one- and two-point correlation functions suffice to fully specify the noise statistical properties. However, the Gaussian assumption needs not be justified a priori and it should rather be validated (or falsified) by the QNS protocol itself. A number of realistic scenarios motivate the consideration of non-Gaussian noise regimes. Statistical processes that are responsible for electronic current fluctuations in mesoscopic devices or the $1/f$ noise ubiquitously encountered in solid-state quantum devices are not Gaussian in general[26]. In superconducting circuits, previous studies have shown that a few two-level defects within Josephson tunnel junctions can interact strongly with the qubit[27-31], the resulting decoherence dynamics showing marked deviations from Gaussian behavior under both free evolution and dynamical decoupling protocols[7,32,33]. More generally, non-Gaussian noise statistics may be expected to arise whenever a qubit is operated outside a linear-response regime, either due to strong coupling to a discrete environment[34] or to a non-linear energy dispersion relationship. The latter feature, which has long been appreciated to influence dephasing behavior at optimal points[35], is common to all state-of-the-art superconducting qubit archetypes[36-39]. Thus, statistical correlations higher than second order and their corresponding multi-dimensional Fourier transforms must be taken into account for complete characterization. From a signal-processing standpoint, this translates into the task of higher-order spectral estimation[40].

In this work, we experimentally demonstrate non-Gaussian QNS by building on the estimation procedure proposed by Norris et al.[41]. While we employ a flux-tunable superconducting qubit as a sensor, our methodology is portable to other physical testbeds in which classical dephasing noise is the dominant decoherence mechanism. We show how non-Gaussianity distinctively modifies the phase evolution of the sensor's coherence, resulting in an observable signature to which the spectrum (or power spectral density, PSD) is completely insensitive and which is instead encoded in the leading higher-order spectrum, the bispectrum. Unlike the original proposal[41], the QNS protocol we introduce here makes use of a statistically motivated maximum likelihood approach. This renders the estimation less susceptible to numerical instability, while allowing measurement errors to be incorporated and both the PSD and the bispectrum to be inferred using a single measurement setup. In order to obtain a clean benchmark for our spectral estimation procedure, we engineer a non-Gaussian noise model by injecting Gaussian flux at the sensor's degeneracy point, resulting in non-Gaussian frequency noise. The noise implementation is validated by verifying the observed power dependence of the leading cumulants against the expected one. Both the reconstructed PSD and the bispectrum are found to be in quantitative agreement with theoretical predictions within error bars.

## Results

**Non-Gaussian dephasing noise.** Before introducing our experimental test bed, we present the general setting to which our analysis is relevant: a qubit sensor evolving under the combined action of non-Gaussian classical dephasing noise and suitably designed sequences of control pulses. By working in an interaction frame with respect to the internal qubit Hamiltonian and the applied control, and letting $\hbar = 1$, the controlled open-system Hamiltonian may be written as $H(t) = y_p(t)B(t)\sigma_z/2$, where $B(t)$ is a stochastic process describing dephasing noise relative to the qubit's eigenbasis defined by the Pauli operator $\sigma_z$. The control switching function $y_p(t)$ accounts for a sequence $p$ of instantaneous $\pi$ rotations about the $x$ or $y$ axis, starting from initial value $y_p(0) = +1$ and toggling between $\pm 1$ with every application of a pulse. Under such a pure-dephasing Hamiltonian, the qubit coherence is quantified by the time-dependent expectation value $\langle\sigma_+(t)\rangle \equiv e^{-\chi(t)+i\phi(t)}\langle\sigma_+(0)\rangle$, where the influence of the noise is captured by the decay and phase parameters $\chi(t)$ and $\phi(t)$. These parameters may be formally expanded in terms of noise cumulants, $C^{(k)}(t_1, \ldots, t_k)$, $k \in \{1, 2, \ldots, \infty\}$, with $\chi(t)$ taking contribution only from even cumulants and $\phi(t)$ only from odd cumulants[41]. Physically, the $k$th-order cumulant is determined by the multi-time correlation functions $\mathbb{E}[B(t_1), \ldots, B(t_j)]$, with $j \leq k$, where $\mathbb{E}[\cdot]$ denotes the ensemble average over noise realizations.

Since the statistical properties of Gaussian noise are entirely determined by one- and two-point correlation functions, cumulants of order $k \geq 3$ vanish identically. By contrast, for non-Gaussian noise, all cumulants can be non-zero in principle. Assuming that noise is stationary, so that the mean of the process $\mathbb{E}[B(t)] = C^{(1)}(0) \equiv \mu_B$ is constant, the phase parameter may be written as $\phi(t) = \mu_B F_p(0, t) + \varphi(t)$, with the Fourier transform $F_p(\omega, t) \equiv \int_0^t ds\, e^{-i\omega s} y_p(s)$ being the fundamental FF associated to the control[12]. This expression separates the phase due to the noise mean, which arises for both Gaussian and non-Gaussian noise, from a genuinely *non-Gaussian phase* $\varphi(t)$, which captures the contribution of all odd noise cumulants with $k \geq 3$. For sufficiently small time or noise strength, we can neglect terms of order $k > 3$ in the cumulant expansion, leading to

$$\chi(t) \approx \frac{1}{2\pi}\int_{\mathbb{R}} d\omega |F_p(\omega, t)|^2 S(\omega), \quad (1)$$

$$\varphi(t) \approx -\frac{1}{3!(2\pi)^2}\int_{\mathbb{R}^2} d\vec{\omega}\, G_p(\vec{\omega}, t) S_2(\vec{\omega}), \quad (2)$$

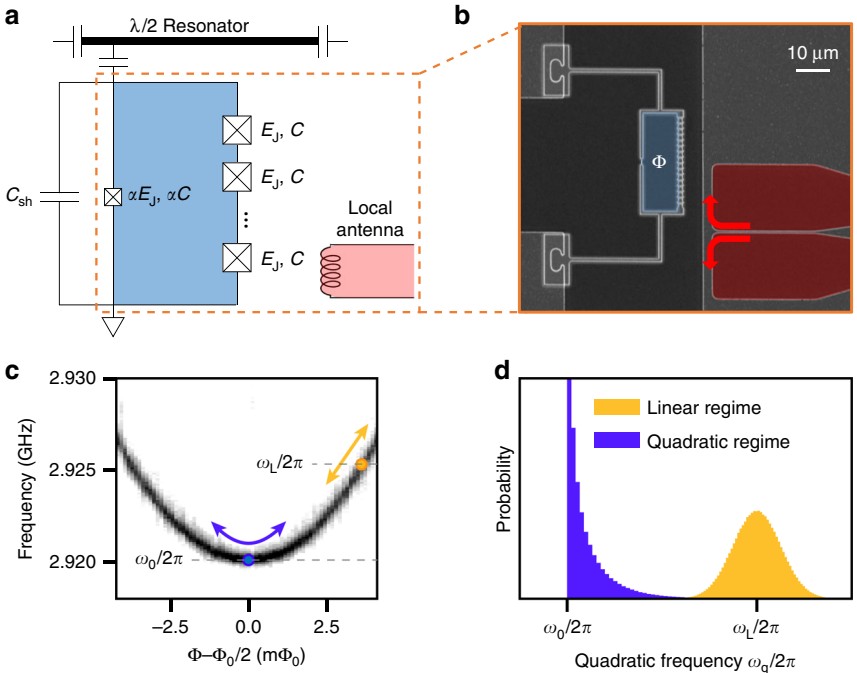

**Fig. 1** Experimental setup and non-Gaussian dephasing noise in a superconducting qubit. **a** Schematic of the circuit QED system. An engineered flux qubit comprises a superconducting loop (blue) interrupted by one small-area and eight large-area Josephson junctions (crosses) and is inductively coupled to a local antenna (red). The qubit junctions have internal capacitance, $C$ and $\alpha C$, and are externally shunted by capacitance $C_{sh}$. See Supplementary Note 1. **b** SEM image of the device. The flux threading the qubit loop $\Phi$ is modulated by applying a current through the local antenna. **c** Frequency spectroscopy of the qubit's $|0\rangle \rightarrow |1\rangle$ transition. At (away from) the degeneracy point $\Phi = \Phi_0/2$, the qubit frequency $\omega_q$ has a quadratic (linear) dependence on the external flux, as indicated by the indigo (yellow) arrow. **d** Probability distribution of the qubit frequency under Gaussian flux noise in the linear regime (yellow) vs. the quadratic regime (indigo). In the quadratic regime, the right-skewness of the distribution illustrates the non-Gaussianity of the resulting noise process

where $\vec{\omega} \equiv (\omega_1, \omega_2)$ and the second and third noise cumulants enter the qubit dynamics through their Fourier transforms: the PSD or spectrum, $S(\omega) \equiv \int_{\mathbb{R}} d\tau e^{-i\omega\tau} C^{(2)}(0, \tau)$, and the second-order polyspectrum or bispectrum, $S_2(\vec{\omega}) \equiv \int_{\mathbb{R}^2} d\vec{\tau} e^{-i\vec{\omega}\cdot\vec{\tau}} C^{(3)}(0, \tau_1, \tau_2)$, with $\vec{\tau} \equiv (\tau_1, \tau_2)$. In the frequency domain, the influence of such spectra is "filtered" by a corresponding generalized FF—in particular, $G_p(\vec{\omega}, t) \equiv F_p(-\omega_1, t) F_p(-\omega_2, t) F_p(\omega_1 + \omega_2, t)$[12]. Since, to leading order, non-Gaussian features arise in our setting from $S_2(\vec{\omega})$, non-Gaussianity of a noise process will be detected and characterized through measurements of $\varphi(t)$.

**Experimental setup and noise validation.** Our circuit quantum electrodynamics (QED) system[42,43] contains an engineered flux qubit[44], which is designed to enable fast single-qubit gates with high fidelity at its flux degeneracy point ($F_g > 99.9\%$; see Supplementary Notes 1 and 2). Single-qubit operations are performed using cosine-shaped microwave pulses, applying an optimal-control technique to suppress leakage to higher levels[45]. Inductive coupling to a local antenna is used to modulate the external flux $\Phi$ threading the qubit loop interrupted by Josephson junctions (Fig. 1a, b). Near the degeneracy (or optimal[35]) point $\Phi = \Phi_0/2$, with $\Phi_0$ the flux quantum, the $|0\rangle \rightarrow |1\rangle$ transition frequency $\omega_q$ has an approximately quadratic dependence on the external flux $\Phi$ (Fig. 1c). Hence, a sufficiently slow time-dependent external flux $\Phi(t)$ enables adiabatic modulation of the qubit frequency, leading to

$$B(t) = \beta_\Phi [\Delta\Phi(t)]^2, \qquad \Delta\Phi(t) \equiv \Phi(t) - \Phi_0/2, \qquad (3)$$

where $\beta_\Phi$ is the quadratic coefficient in the dispersion relation between qubit frequency and flux. Crucially, any non-linear function of a Gaussian process leads to non-Gaussian noise. In

particular, the quadratic function implemented in Eq. (3) transduces zero-mean Gaussian flux noise into non-Gaussian qubit-frequency noise (Fig. 1c, d). Assuming that the noise is entirely contributed by the applied $\Delta\Phi(t)$, and that $S_\Phi(\omega)$ denotes the corresponding PSD, the mean $\mu_B$, PSD $S(\omega)$, and bispectrum $S_2(\omega_1, \omega_2)$ of $B(t)$ are, respectively, given by

$$\mu_B = \frac{\beta_\Phi}{2\pi} \int_{\mathbb{R}} d\omega S_\Phi(\omega), \qquad (4)$$

$$S(\omega) = \frac{\beta_\Phi^2}{\pi} \int_{\mathbb{R}} du S_\Phi(u) S_\Phi(\omega - u), \qquad (5)$$

$$S_2(\omega_1, \omega_2) = \frac{4\beta_\Phi^3}{\pi} \int_{\mathbb{R}} du S_\Phi(u) S_\Phi(\omega_1 + u) S_\Phi(\omega_2 - u). \qquad (6)$$

In the experiment, we choose $S_\Phi(\omega)$ to be a zero-mean Lorentzian function, $S_\Phi(\omega) = (P_0/\pi\omega_c)/[1 + (\omega/\omega_c)^2]$, where $\omega_c/2\pi$ (=0.5 MHz) and $P_0$ denote the cutoff frequency and the power of the applied flux noise, respectively. As is apparent from Eqs. (4)–(6), cumulants of order $k = 1$, 2, and 3 are distinguished by their linear, quadratic, and cubic dependence on power, respectively.

We first validate the intended engineered non-Gaussian noise by demonstrating consistency of the measured power dependence of $\chi$ and $\phi$ with the above prediction. The qubit is initialized to the $+y$ axis by applying a $\pi/2$ pulse about $x$ (rotation $R_x(\pi/2)$), and Gaussian flux noise is injected while it evolves in the $xy$ plane of the Bloch sphere for time $T$. During this evolution, we apply a Carr–Purcell–Meiboom–Gill (CPMG) sequence consisting of two refocusing $\pi$ pulses about $y$ (Fig. 2a). At the end of this sequence ($t = T$), the effect of the first cumulant of the noise cancels out ($F_p(0, T) = 0$) and, as a result, the measured phase becomes solely determined by odd cumulants of order $k \geq 3$: $\phi(T) = \varphi(T)$. To

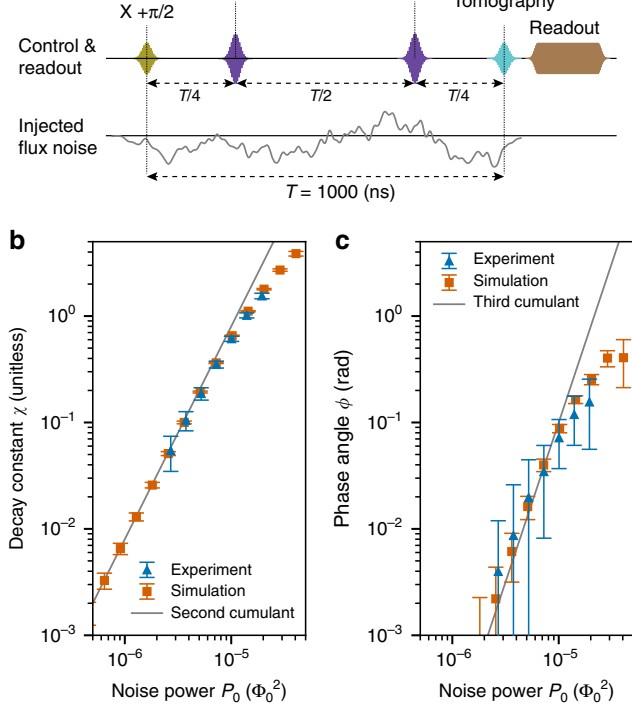

**Fig. 2** Power dependence of decay constant ($\chi$) and phase angle ($\phi$). **a** Pulse scheme for measuring the power dependence of $\chi$ and $\phi$, consisting of a CPMG sequence of length $T = 1\,\mu s$ with two $\pi$ pulses. Flux noise waveforms are temporally tailored to affect the qubit only while it evolves on the transverse plane. **b** Decay constant $\chi = -\log\left(\sqrt{\langle\sigma_x\rangle^2 + \langle\sigma_y\rangle^2}\right)$ and **c** phase angle $\phi = \tan^{-1}(-\langle\sigma_x\rangle/\langle\sigma_y\rangle)$ at time $t = T$, after application of a CPMG sequence as a function of the applied noise power $P_0$. A cubic power dependence of $\phi$, for sufficiently weak noise, corroborates non-Gaussianity of the engineered noise. Error bars represent 95% confidence intervals

estimate both $\phi$ and $\chi$, we measure $\langle\sigma_x\rangle$ and $\langle\sigma_y\rangle$ by applying appropriate tomography pulses at time $t = T$, before readout in the $\sigma_z$-basis.

Figure 2b, c shows $\chi$ and $\phi$ as a function of injected flux noise power $P_0$ for both the experiment (blue triangles) and Monte Carlo simulations accounting for all cumulants of the applied noise (orange squares, see Supplementary Note 5). Substituting Eqs. (5) and (6) into Eqs. (1) and (2), we also plot the resulting ideal weak-power behavior (gray solid) considering only the leading-order cumulants of order two and three for $\chi$ and $\phi$, respectively. For sufficiently small $P_0$, these ideal values are in good agreement with data from both experiment and simulation, showing that $\chi$ and $\phi$ obey the quadratic and cubic power dependences that are expected for the square of a Gaussian flux-noise process under the CPMG sequence. In particular, the cubic dependence of $\phi$ at small $P_0$ corroborates the presence of a non-zero third-order cumulant, which would not exist for Gaussian noise. Deviations of the simulations and experimental data from the ideal behavior at large $P_0$ are attributable to the contribution of cumulants of order $k > 3$. The quantitative agreement between theory, experiment, and simulation observed at low power demonstrates our capability to produce and sense engineered noise that dominates over native one over the relevant parameter regime and exhibits well-controlled cumulants, a necessary first step in the experimental validation of non-Gaussian QNS.

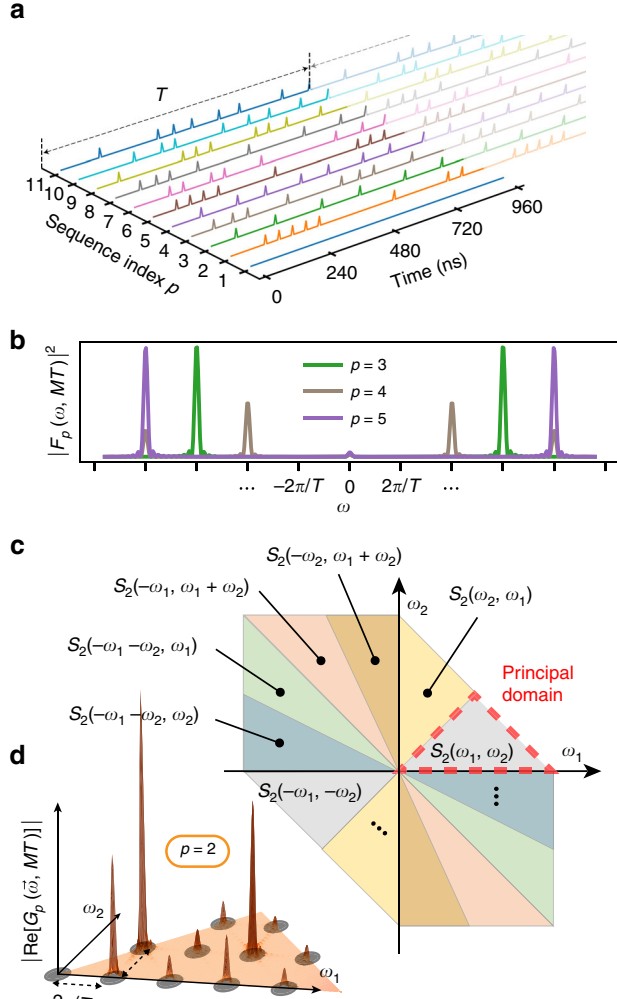

**Fig. 3** A protocol for non-Gaussian noise spectroscopy. **a** Timing diagrams of control pulse sequences. The length of the base sequence is $T = 960$ ns, $p = 1$ corresponds to a single free-evolution period, whereas sequences $p = 2, \ldots, 11$ are repeated $M = 10$ times. Only $\pi$-pulses are shown and all $\pi$-pulses are around the $y$ axis (see Supplementary Note 4 for details). **b** $|F_p(\omega, MT)|^2$ for $p = 3, 4, 5$ as a function of angular frequency $\omega$. **c** Symmetries of the bispectrum of a classical stationary noise process. **d** 2D grid representing the harmonic frequencies (black circles) in the principal domain $\mathcal{D}_2$ (orange area) in which the bispectrum is sampled. The amplitude of the relevant contribution of the FF in $\mathcal{D}_2$, $|\text{Re}[G_p(\vec{\omega}, MT)]|$, for $p = 2$, (red surface plot) is shown on the top of the grid

**Non-Gaussian noise spectroscopy.** Having established that $\chi$ and $\phi$ follow their expected behavior, we move on to fully characterizing the first three cumulants of our engineered noise source by measuring its mean, PSD, and bispectrum. Since the noise mean, $\mu_B$, manifests itself through a qubit-frequency shift, it can be measured from a simple parameter estimation scheme based on Ramsey interferometry. By contrast, we aim to perform a non-parametric estimation of both the PSD and bispectrum, that is, to reconstruct them at a set of discrete points in frequency space without assuming a prior functional form. Figure 3 illustrates our protocol for simultaneous estimation of the PSD and bispectrum, in which filter design—the selection of pulse times in a control sequence so that the corresponding FF has a particular shape—is instrumental. Building on ref. [13], applying $M \gg 1$ repetitions of a "base" pulse sequence $p \in \{1, 2, \cdots, P\}$, with duration $T$, shapes the FF $|F_p(\omega, MT)|^2$ into a frequency comb with narrow teeth probing

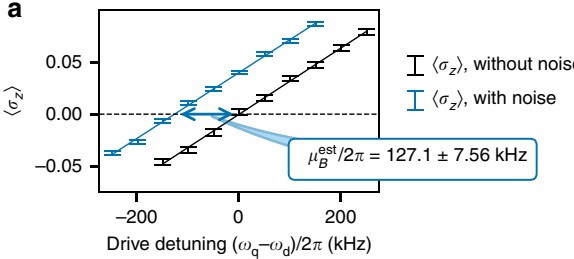

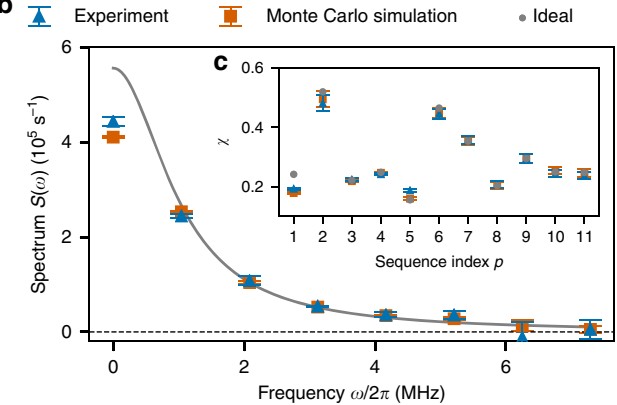

**Fig. 4** Gaussian spectral estimation: noise mean and PSD. **a** Measured values of $\langle \sigma_z \rangle$ after a 50-ns-long Ramsey sequence vs. drive detuning $D = \omega_q - \omega_d$. The separation between the x-intercepts of the two fitted lines gives the mean $\mu_B^{est}$ of the injected dephasing noise. **b** Comparison of the experimental reconstruction (blue triangle) and Monte Carlo simulation (orange square) with the ideal PSD (gray solid line). **c** Decay constants $\chi$. Except for $p = 1$, the ideal data (gray circles) are in very good agreement with both the experimental results and Monte Carlo simulations. Error bars represent 95% confidence intervals

$S(\omega)$ at harmonics $k\omega_h$, with $k$ an integer and $\omega_h \equiv 2\pi/T$ (Fig. 3a, b). This result generalizes to filters relevant to higher-order spectra[12]: under sequence repetition, $G_p(\vec{\omega}, MT)$ becomes a two-dimensional (2D) "hyper-comb" with teeth probing $S_2(\vec{\omega})$ at $\vec{\omega} \in \{\vec{k}\omega_h\}$, where $\vec{k} \equiv (k_1, k_2)$ with $k_1$ and $k_2$ integers (Fig. 3d).

For both the PSD and bispectrum, distinct pulse sequences have the effect of giving different weights to the comb teeth, granting access to complementary information about $S(k\omega_h)$ and $S_2(\vec{k}\omega_h)$, enabling their reconstruction. More specifically, in both cases, the basic steps of our protocol consist of (i) applying a set of sufficiently distinct pulse sequences $p$ (Fig. 3a); (ii) measuring the corresponding decay and phase parameters; and (iii) solving the resulting systems of linear equations, which give $\chi_p(MT)$ and $\varphi_p(MT)$ as a function of $S(k\omega_h)$ and $S_2(\vec{k}\omega_h)$. Since classical noise has a spectrum with even symmetry, $S(\omega) = S(-\omega)$, the PSD is specified across all frequency space by its values at positive frequencies. Likewise, the bispectrum is completely specified by its values over a subspace $\mathcal{D}_2$ known as the principal domain[41,46], illustrated in Fig. 3c. Reconstructing the bispectrum over $\mathcal{D}_2$ and exploiting the symmetries that $S_2(\vec{\omega})$ exhibits (shown in Fig. 3c) thus suffices to retrieve the bispectrum over the whole relevant frequency domain.

Figure 4 presents experimental results for determining the mean and PSD, which suffice to characterize the noise process in the Gaussian approximation. To measure $\mu_B$ by Ramsey interferometry, we apply a pair of $\pi/2$ pulses with a drive at frequency $\omega_d$, first about $x$ at time $t = 0$ ($R_x(\pi/2)$), and then about $y$ at time $t = T$ ($R_y(\pi/2)$). We choose a pulse interval $T = 50$ ns, which is short enough for cumulants of order higher than one to be negligible,

but long enough to avoid pulse overlap. The qubit polarization at time $t_f$ after the two pulses is then $\langle \sigma_z(t_f) \rangle \approx (D + \mu_B)T'$, where $D \equiv \omega_q - \omega_d$ is the drive detuning, and $T'$ is an effective time interval that accounts for the finite-width pulse shape (see Supplementary Note 6). Thus, plotting $\langle \sigma_z(t_f) \rangle$ as a function of $D$ produces a straight line whose x-intercept is $-\mu_B$, leading to an estimate that is insensitive to the pulse shape to first order in the cumulant expansion. Figure 4a presents data for measurements of $\langle \sigma_z(t_f) \rangle$, and shows how we isolate the contribution of the engineered noise source by performing the sequence with (blue data set) and without (black data set) applied noise. The mean of the engineered noise is estimated by subtracting the x-intercepts of the straight lines that are fitted to each data set. Performing these fits under the conditional normal model of linear regression (see Supplementary Note 6) yields the estimate $\mu_B^{est}/2\pi = 127.1 \pm 7.56$ kHz, where the uncertainty corresponds to the 95% confidence interval calculated from the asymptotic normal distribution of qubit polarization.

To estimate the PSD by the comb approach outlined above, we use both a period of free evolution ($p = 1$) and $M = 10$ repetitions of base sequences $p = 2, ..., 11$ illustrated in Fig. 3a (see Supplementary Note 4 for the actual pulse times). For $M \gg 1$, the FF entering the decay constant in Eq. (1) becomes approximately $|F_p(\omega, MT)|^2 \approx \frac{M}{T}|F_p(\omega, T)|^2 \sum_{k=-\infty}^{\infty} \delta(\omega - k\omega_h)$, which enables us to sample the PSD at the harmonic frequencies in terms of the (known) control FFs,

$$\chi_p(MT) \approx \frac{M}{T} \sum_{k \in \mathcal{K}_1} |F_p(k\omega_h, T)|^2 S(k\omega_h). \tag{7}$$

Here, we have used the even symmetry of the PSD, and the high-frequency decay of the PSD and FFs to truncate the comb to a finite set of positive harmonics, $\mathcal{K}_1 \equiv \{0, ..., K-1\}$. Rather than solving the above linear system by matrix inversion as in ref. [13], we employ a statistically motivated maximum-likelihood estimate (MLE), which takes experimental error into account (see Supplementary Note 7). Using measurements of $\chi_p(MT)$ for each of the same $P = 11$ control sequences to be used for the bispectrum estimation, we find a well-conditioned system for $K = 8$.

Figure 4b compares the experimentally estimated PSD at the $K = 8$ harmonics (blue triangles) with the ideal PSD obtained from Eq. (5) for our engineered noise (solid gray line) and Monte Carlo simulations of the QNS protocol (orange squares). The experimental and simulated estimates of the PSD are plotted along with 95% confidence intervals obtained from the asymptotic normal distribution of the decay constants. Figure 4c shows the experimental and simulated values of $\chi_p(MT)$ that were used as input for the reconstructions, along with ideal values obtained by substituting Eq. (5) into Eq. (1) and approximating the FF by the ideal (infinite) comb as given above. The PSD is slightly underestimated at zero frequency in both the experiment and Monte Carlo simulation since the FF of sequence $p = 1$ (a 960-ns-long free induction decay) is comparable in bandwidth to the PSD, whereas the reconstruction procedure assumes the PSD is sampled by infinitely narrow FFs. The disagreement of the experimental and simulated $\chi_p(MT)$ for $p = 1$ with the ideal value is also explained by the non-negligible bandwidth of the FF (Fig. 4c). Apart from these well-understood discrepancies at $\omega = 0$, the quantitative agreement of the experimental reconstruction with simulations and ideal values is remarkable, which demonstrates that our protocol is able to reliably characterize Gaussian features of the applied noise.

We are now in a position to present our key result: the reconstruction of the noise bispectrum. As anticipated, this entails

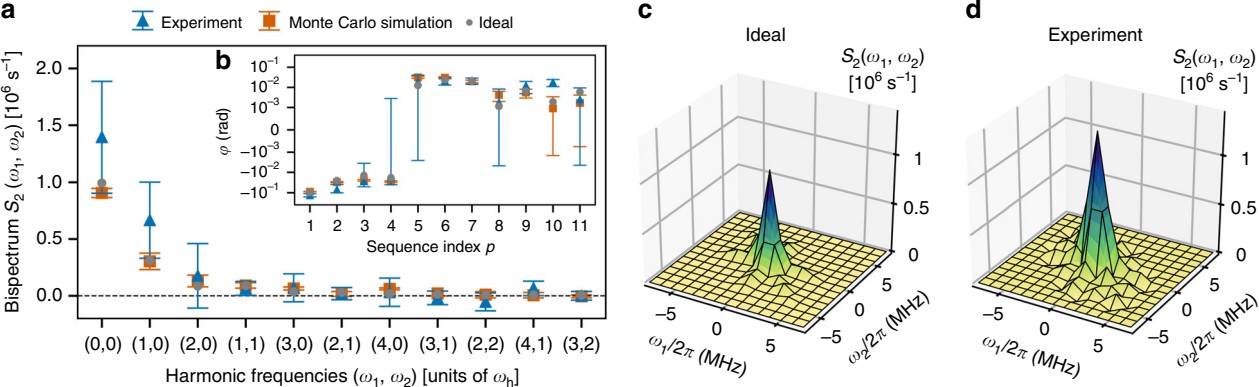

**Fig. 5 Non-Gaussian spectral estimation: noise bispectrum. a** Experimental data (blue triangles), Monte Carlo simulations (orange squares), and ideal values (gray circles) for the bispectrum of the engineered dephasing noise. The error bars indicate that the experimental bispectrum agrees with both the ideal bispectrum and Monte Carlo simulations of the protocol within 95% confidence intervals. **b** Estimated non-Gaussian phase angles φ. Error bars represent 95% confidence intervals. **c** 3D visualization of the ideal bispectrum. **d** 3D visualization of the reconstructed bispectrum for the experimental data

a higher-dimensional analog of the comb-based approach used for the PSD. We estimate the non-Gaussian phase given in Eq. (2) by subtracting the contribution of the noise mean from the total measured phase, $\varphi_p(MT) = \phi_p(MT) - \mu_B F_p(0, MT)$, where we replace $\mu_B$ by $\mu_B^{\text{est}}$ experimentally determined above. After $M \gg 1$ repetitions of sequence $p$, the FF becomes a 2D comb (Fig. 3d), and the non-Gaussian phase becomes a sampling of the bispectrum at the harmonics $\vec{k}\omega_h$, that is,
$\varphi_p(MT) = -\frac{M}{3!T^2} \sum_{\vec{k} \in \mathbb{Z}^2} G_p(\omega_h\vec{k}, T) S_2(\omega_h\vec{k})$. Since both the filter
and bispectrum decay at high frequencies, we can truncate this sum to a finite number of $\vec{k} = (k_1, k_2)$. As the bispectrum is completely specified by its values on the principal domain, we may further restrict our consideration to a subset of harmonics, $\mathcal{K}_2 \equiv \{\vec{k}_1, \dots, \vec{k}_N\} \subset \mathcal{D}_2$ (Fig. 3d). The non-Gaussian phase then becomes

$$\varphi_p(MT) = -\frac{M}{3!T^2} \sum_{\vec{k} \in \mathcal{K}_2} m(\omega_h\vec{k}) \text{Re}[G_p(\omega_h\vec{k}, T)] S_2(\omega_h\vec{k}), \quad (8)$$

where the multiplicity $m(\omega_h\vec{k})$ accounts for the number of points equivalent to $S_2(\omega_h\vec{k})$ by the symmetry properties of the bispectrum. Also on account of these symmetries, the imaginary component of $G_p(\omega_h\vec{k}, T)$ cancels when the sum is restricted to $\mathcal{D}_2$ (see Supplementary Note 8).

By measuring the non-Gaussian phase for $P \geq N$ different control sequences, we can construct a vector $\vec{\varphi} = [\varphi_1(MT), \dots, \varphi_P(MT)]^T$ and a linear system of the form

$$\vec{\varphi} = \mathbf{A}\vec{S}_2, \quad \mathbf{A}_{pn} = -\frac{M}{3!T^2} m(\omega_h\vec{k}_n) \text{Re}[G_p(\omega_h\vec{k}_n, T)], \quad (9)$$

where $\vec{S}_2 = [S_2(\omega_h\vec{k}_1), \dots, S_2(\omega_h\vec{k}_N)]^T$ contains the bispectrum at the harmonics in $\mathcal{K}_2$ and $\mathbf{A}$ is a $P \times N$ reconstruction matrix. The simplest way to estimate the bispectrum from this linear system is the least-squares estimate employed in ref. [41], involving the (pseudo-)inverse of the reconstruction matrix, $\vec{S}_2^{\text{est}} = \mathbf{A}^{-1}\vec{\varphi}$. As in the case of PSD estimation, a potential drawback of this inversion-based approach is numerical instability stemming from an ill-conditioned $\mathbf{A}$, which occurs when the FFs have a high degree of spectral overlap. Since ill-conditioning makes the least-squares estimate sensitive to even small errors in the measured phases, we again utilize a maximum-likelihood approach with

optional regularization to further increase stability (see Supplementary Note 8). From the asymptotic Gaussian distribution of the measurement outcomes of $\vec{\varphi}$, the regularized maximum-likelihood estimate (RMLE) is found as

$$\vec{S}_2^{\text{RMLE}} = \underset{\vec{S}_2}{\text{argmin}} \left[ \frac{1}{2}(\mathbf{A}\vec{S}_2 - \vec{\varphi})^T \mathbf{\Sigma}^{-1}(\mathbf{A}\vec{S}_2 - \vec{\varphi}) + \left\| \lambda \mathbf{D}\vec{S}_2 \right\|_2^2 \right],$$
$$(10)$$

where $\|\cdot\|_2$ denotes the $L_2$-norm and $\lambda \geq 0$ parametrizes the strength of the regularization[47]. Due to its dependence on the covariance matrix $\mathbf{\Sigma}$, the RMLE down-weights phase measurements with larger error. Numerical stability is increased by the regularizer $\left\| \lambda \mathbf{D}\vec{S}_2 \right\|_2^2$, which acts as an effective constraint. When the smoothing matrix $\mathbf{D}$ is proportional to $\mathbf{I}$, the regularizer reduces to the well-known Tikhonov (or $L_2$) form. Since the numerical stability afforded by regularization comes at the cost of additional bias, choosing the regularization strength is a nontrivial task. In Supplementary Note 8, we detail how we have selected $\lambda$ based on the so-called "L-curve criterion". Interestingly, since $\mathbf{A}$ is sufficiently well-conditioned for the sequences we have chosen, we find that regularization gives negligible benefit. Accordingly, we use $\lambda = 0$ (which recovers standard MLE) in our experimental reconstructions.

Figure 5a compares the results of the non-Gaussian spectral estimation for the harmonics in the principal domain for the experiment (blue triangles) with both the ideal bispectrum obtained from Eq. (6) (gray circles) and from Monte Carlo simulations (orange squares). To estimate the experimental bispectrum, we input the measured data for $\vec{\varphi}$ and $\mathbf{\Sigma}$ shown in Fig. 5b into $\vec{S}_2^{\text{RMLE}}$ given by Eq. (10). The ideal values of $\varphi_p$, also shown in Fig. 5b, are obtained by substituting Eq. (6) into Eq. (2). We further display 3D representations of the full bispectra, obtained by applying relevant symmetries to the data on $\mathcal{D}_2$, for the ideal (Fig. 5c) and experimental (Fig. 5d) cases, respectively. Ignoring error bars, the reconstructed bispectrum appears to be an overestimate with respect to the ideal one. This error may be attributed to noise during the finite-duration control pulses used in the experiment, leading to effective pulse infidelity. Upon taking the error bars in Fig. 5a into consideration, however, the ideal and simulated values of the bispectrum lie within the 95% confidence intervals of the experimental reconstruction, suggesting that this estimation error is statistically insignificant and thus

successfully extending the validation of our QNS protocol to the leading non-Gaussian noise cumulant.

Although the theoretical bispectrum falls within the 95% confidence interval of the estimate, reducing the magnitude of uncertainties is clearly necessary to push the application of non-Gaussian QNS to uncontrollable native noise, whose strength may be comparatively weak. We note that the spectral characterization of the non-Gaussian noise process engineered in this experiment requires an extremely precise estimation of $\mu_B$. Since reconstructions of the bispectrum are obtained using $\varphi_p(MT) = \phi_p(MT) - \mu_B F_p(0, MT)$, the uncertainty in $\mu_B^{est}$ propagates to $\varphi_p(MT)$ when $p$ has zero filter order, i.e. $F_p(0, MT) \neq 0$. These sequences play a crucial role in estimating the bispectrum at the "zero points", grid points $(\omega_1, \omega_2)$ with $\omega_1 = 0$ or $\omega_2 = 0$. Since $\mu_B$ is much larger than the third cumulant for the current noise process, even a small relative uncertainty in $\mu_B^{est}$ can lead to greater error in the bispectrum estimate at the zero points, as the error bars in Fig. 5a attest.

## Discussion

In summary, we experimentally demonstrated high-order spectral estimation in a quantum system. By producing and sensing engineered noise with well-controlled cumulants, we were able to successfully validate a spectroscopy protocol that reconstructs both the PSD and the bispectrum of non-Gaussian dephasing noise. Our theory and experimental demonstration lay the groundwork for future research aiming at complete spectral characterization of realistic non-Gaussian noise environments in quantum devices and materials. Theoretically, we expect that the regularized maximum-likelihood estimation approach to QNS we invoked here will prove crucial to ensure stable spectral reconstructions in more general settings. Devising alternative estimation protocols based on optimally band-limited control modulation and multitaper techniques[48] appears especially compelling, in view of recent advances in the Gaussian regime[24,49]. We believe that obtaining a complete spectral characterization will ultimately provide deeper insight into the physics and interplay of different microscopic noise mechanisms, including non-classical non-Gaussian noise, as possibly arising from photon-number-mediated non-linear couplings[50].

## Data availability

The data that support the findings of this study may be made available from the corresponding authors upon request and with the permission of the US Government sponsors who funded the work.

## Code availability

The code used for the analyses may be made available from the corresponding authors upon request and with the permission of the US Government sponsors who funded the work.

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

## Acknowledgements

It is a pleasure to thank K. Harrabi, M. Kjaergaard, P. Krantz, G. A. Paz-Silva, J. I. J. Wang, and R. Winik for insightful discussions, and M. Pulido for generous assistance. We thank B. M. Niedzielski for the SEM image of the device. This research was funded by the U.S. Army Research Office grant No. W911NF-14-1-0682 (to L.V. and W.D.O.); and by the Department of Defense via MIT Lincoln Laboratory under Air Force Contract No. FA8721-05-C-0002 (to W.D.O.). Y.S. and F.B. acknowledge support from the Korea Foundation for Advanced Studies and from the Fonds de Recherche du Québec – Nature et Technologies, respectively. The views and conclusions contained herein are those of the authors and should not be interpreted as necessarily representing the official policies or endorsements, either expressed or implied, of the U.S. Government.

## Author contributions

Y.S., F.Y., and S.G. performed the experiments. F.B and Y.S. carried out numerical simulations and analyzed the data, and L.V., L.M.N., S.G., and W.D.O. provided feedback. L.M.N., F.B., and L.V. designed the pulse sequences and developed the estimation protocol used in the experiment. F.Y. and S.G. designed the device and D.K.K. and J.L.Y. fabricated it. J.Y.Q. and U.L. provided experimental assistance. Y.S., F.B., L.M. N., and L.V. wrote the manuscript with feedback from all authors. L.V., S.G., T.P.O., and W.D.O. supervised the project.

## Additional information

**Competing interests:** The authors declare no competing interests.

