## [Peer Review File · Nature Communications]

Reviewers' comments:

Reviewer #1 (Remarks to the Author):

The major claim of the manuscript is the first experimental validation of a quantum control protocol, proposed in reference 36, allowing for reconstruction of spectrum and bispectrum of engineered non-Gaussian classical dephasing noise acting on a superconducting flux qubit sensor. This is indeed a relevant achievement toward spectral characterization of realistic non-Gaussian noise in quantum devices. In fact, so far experimental quantum noise spectroscopy relied on the assumption that the target noise is Gaussian. Reported results are a key step to fully exploit the potential of quantum technologies.

Results are technically sound and clearly presented. The theoretical analysis is also based on refs. 12 and 13, but applying a statistically motivated maximum likelihood procedure for the power spectrum estimate in order to take experimental errors into account. Theoretical details are clearly illustrated in the Supplementary Information. The detail is sufficient to reproduce the statistical analysis.

I suggest the paper for publication in Nature Communications after revision addressing the following recommendations:

1) It would be worth to comment on the "real" noise acting on the flux qubit sensor and how it combines/compares with engineered noise. In fact, even when operating at the degeneracy point, the sensor will be subject to non Gaussian noise due to Gaussian magnetic flux noise, as discussed in Yuriy Makhlin and Alexander Shnirman Phys. Rev. Lett. 92, 178301, 2004.

2) It would be appropriate to refer to, or at least mention, other existing literature addressing either different methods of employing qubits as sensors of their own non-Gaussian noise or pointing out deviation from Gaussian behaviour due to strong coupling of qubits with two level defects. A partial list is:

G. Falci, A. D'Arrigo, A. Mastellone, and E. Paladino Phys. Rev. A 70, 040101(R), 2004

Tero T. Heikkilä, Pauli Virtanen, Göran Johansson, and Frank K. Wilhelm Phys. Rev. Lett. 93, 247005, 2004

Y. M. Galperin, B. L. Altshuler, J. Bergli, D. Shantsev, and V. Vinokur Phys. Rev. B 76, 064531, 2007

Tatsuro Yuge, Susumu Sasaki, and Yoshiro Hirayama, Phys. Rev. Lett. 107, 170504, 2011

V. Zaretsky, B. Suri, S. Novikov, F. C. Wellstood, and B. S. Palmer Phys. Rev. B 87, 174522, 2013

Lisenfeld et al Scientific Reports volume 6, 23786, 2016.

3) Supplementary Information, equations (30), (32), (33): the LHS apparently should not depend on time whereas quantities on the RHS do have an explicit time dependence. Please clarify.

4) Abstract, it might be unclear to non specialized readers what a "sparse environment" is. I suggest modifying the sentence.

Reviewer #2 (Remarks to the Author):

The characterization of noise by a qubit as a sensor is quite important to the development of quantum information science, namely in the development of quantum sensing and quantum computing processors which must contend with noisy environments. There is also no doubt that non-Gaussian noise plays an important role in circuits in general. Examples of this include the ubiquitous $1/f$ noise and noise from two-level systems, which can dominate.

Since non-Gaussian noise must be understood much better for quantum information science, spectral characterization related to new theory work by Viola is quite interesting as a basis of this experimental study in the group led by Oliver. They study input Gaussian noise, but the energy dispersion relationship for the qubit changes the noise, because the qubit frequency is quadratic in

input noise. This creates a non-Gaussian noise which is visible in the qubit phase. Estimated noise in a parameter χ relates to the Gaussian spectral density obtainable from a second order cumulant, and a filter function related to the qubit pulse sequence. This study importantly goes beyond this for the bispectrum which is expressible by a third order cumulant and a higher-order filter function. Alternatively for this setup, the noise can be described as a function of the input PSD and various frequencies related to the input filter function frequencies. The bispectrum has peaks in frequency related to the inverse period in a base sequence, which is 1 MHz ($=1/\text{microsecond}$).

By completing the estimation of noise and verifying that the non-Gaussian noise agrees quantitatively with calculations, the authors have completed a pioneering technique for studying non-Gaussian noise. There is a good possibility that this study of non-Gaussian noise will also lead to a better understanding of non-Gaussian quantum noise, which should be present in quantum sensitive noise phenomena. Therefore further studies with this procedure are expected. The manuscript is important and generally complete. However, I have one concern listed below about the qubit type that I would like to see addressed before publication.

Why is this called a flux qubit when it looks like a fluxonium (Fig 1a)? Is it the Capacitive shunt? Reference 39 is cited, but it only specified that a manuscript on this optimized qubit is "in preparation." It would be nice if the authors could share a preprint of the manuscript because it is important for the authors to convince the reviewing team that the qubit is closer to a flux qubit than a fluxonium. After all, this might be a reasonable assumption since it is also shunted with a chain of JJs like a fluxonium. Of course some parameters are listed, it is just that a choice of parameters could be listed in the supplemental with a justification of why the authors describe it as a flux qubit. Also, the group has previously studied a C-shunted qubit and it would be helpful to compare this qubit to that one. Both changes could be described without describing the method by which the qubit was obtained (which is presumably the subject of the unreleased manuscript). It would be unfortunate if confusion over the qubit type held up scientific verification of the bispectrum estimation reported on here. As a minor point, I believe the bispectrum $S_2(\omega)$ on line 156 needs a vector symbol over the ω .

Re: NCOMMS-19-08909-T

Response to Reviewer #1

We are very pleased with the positive assessment and recommendation of the Referee, in particular his/her appreciation of the key importance of our achievement toward fully exploiting the potential of quantum technologies. We have carefully considered all the comments and recommendations and implemented a number of modifications in the revised version of both the manuscript and supplement. Specifically, we provide point-by-point responses in what follows.

1) It would be worth to comment on the “real” noise acting on the flux qubit sensor and how it combines/compares with engineered noise. In fact, even when operating at the degeneracy point, the sensor will be subject to non Gaussian noise due to Gaussian magnetic flux noise, as discussed in Yuriy Makhlin and Alexander Shnirman Phys. Rev. Lett. 92, 178301, 2004.

This is indeed an important point, on which we had been insufficiently explicit: of course, any Gaussian noise (engineered and “real” i.e., native to the device) is in fact transduced to non-Gaussian qubit frequency noise, by virtue of the non-linear dispersion in Eq. (3). As to the paper the Referee points out, that is indeed what inspired consideration of a quadratic dispersion in our initial theoretical analysis of non-Gaussian noise spectroscopy, in Ref. 42. This important paper has now also been added in the revised version (Ref. 36) and quoted in the introduction. Furthermore, we have explicitly noted in the text [lines 182—186] that Eqs. (4), (5), and (6) hold under the assumption that the relevant noise is solely due to the engineered one; such an assumption is validated *a posteriori*, by verifying agreement with the predicted power dependencies of the decay and phase parameters, reported in Fig. 2. Accordingly, the fact that native noise is effectively negligible over the relevant parameter regime is noted in the revised text [lines 230-231].

2) It would be appropriate to refer to, or at least mention, other existing literature addressing either different methods of employing qubits as sensors of their own non-Gaussian noise or pointing out deviation from Gaussian behaviour due to strong coupling of qubits with two level defects. A partial list is:

We appreciate the importance of placing our contribution in a broader context and thank the Referee for bringing relevant references to our attention.

- *G. Falci, A. D'Arrigo, A. Mastellone, and E. Paladino Phys. Rev. A 70, 040101(R), 2004*
- *Y. M. Galperin, B. L. Altshuler, J. Bergli, D. Shantsev, and V. Vinokur Phys. Rev. B 76, 064531, 2007*

Both of these references have been added (Refs. 33, 34), alongside with Ref. 7, in the introduction [lines 74-75].

- *Tatsuro Yuge, Susumu Sasaki, and Yoshiro Hirayama, Phys. Rev. Lett. 107, 170504, 2011*

This reference was already cited in the original version, and we have kept it (Ref. 10 in the revised version).

- *V. Zaretsky, B. Suri, S. Novikov, F. C. Wellstood, and B. S. Palmer Phys. Rev. B 87, 174522, 2013*
- *Lisenfeld et al Scientific Reports volume 6, 23786, 2016*

Both of these papers have now been appropriately cited in the introduction, Refs. 30 and 32, respectively.

- *Tero T. Heikkilä, Pauli Virtanen, Göran Johansson, and Frank K. Wilhelm Phys. Rev. Lett. 93, 247005, 2004*

While this paper is very interesting and clearly relevant in the context of characterizing non-Gaussian noise signatures, we have opted not to include it because it does not directly address non-Gaussian noise in a qubit device – but, it rather focuses on transport properties in a Josephson junction. As such, we struggled to find a natural and concise way to discuss it in the text. We fully concur with the Referee, however, on the fact that deviations from non-Gaussian statistics have long been appreciated in mesoscopic physics (e.g., as manifest in work on full counting statistics), and that different methods have been used to detect non-Gaussian behavior – albeit, to the best of our knowledge, not resulting in a fully frequency-resolved high-order spectral reconstruction as we achieve. With this in mind, we have instead added a review paper (Ref. 26) pointing to current fluctuations in mesoscopic devices in the introduction [lines 67-68].

3) Supplementary Information, equations (30), (32), (33): the LHS apparently should not depend on time whereas quantities on the RHS do have an explicit time dependence. Please clarify.

We have double-checked the relevant equations, and they are correct as given. Both the left-hand and right-hand sides of Eqs. (30), (32) and (33) do depend on time. Indeed, note that the quantities σ_i^{est} and μ_i ($i=x, y$), defined respectively as the estimated and true expectation values

of the Pauli operator σ_i at time t , depend implicitly on time. In order to simplify notation, however, we have avoided an explicit time argument. For clarification, we have now added sentences beneath Eqs. (23) and (29), emphasizing the time dependence of these quantities.

4) *Abstract, it might be unclear to non specialized readers what a “sparse environment” is. I suggest modifying the sentence.*

We agree with the Referee. We have now replaced the word “sparse” with “discrete”, both in the abstract and introduction [lines 10; 77], as that is both clearer and in line with use in relevant literature (e.g., Refs. 32, 34).

Response to Reviewer #2

We thank the Referee for the positive recommendation. We are glad that he/she appreciates the importance and novelty of our contribution, and its potential for our technique to lead to a deeper understanding of non-Gaussian quantum noise. The Referee has, however, a main concern in regard to the qubit device we use, which he/she would like to see addressed before publication. More specifically:

Why is this called a flux qubit when it looks like a fluxonium (Fig 1a)? Is it the Capacitive shunt? Reference 39 is cited, but it only specified that a manuscript on this optimized qubit is “in preparation.” It would be nice if the authors could share a preprint of the manuscript because it is important for the authors to convince the reviewing team that the qubit is closer to a flux qubit than a fluxonium. After all, this might be reasonable assumption since it is also shunted with a chain of JJs like a fluxonium. Of course some parameters are listed, it is just that a choice of parameters could be listed in the supplemental with a justification of why the authors describe it as a flux qubit. Also, the group has previously studied a C-shunted qubit and it would be helpful to compare this qubit to that one. Both changes could be described without describing the method by which the qubit was obtained (which is presumably the subject of the unreleased manuscript). It would be unfortunate if confusion over the qubit type held up scientific verification of the bispectrum estimation reported on here.

These are important points, on which we are happy to elaborate. The qubit we used here is a “generalized (engineered) flux qubit” -- described in more detail below -- which operates in the capacitively shunted flux-qubit regime and not in the fluxonium regime.

There are two key points to make here:

- 1) Although our engineered flux qubit has an array of junctions, the number of these array junctions, the choice of design parameters, and their role in the circuit are completely

different from fluxonium [1-3]. Since our qubit does not operate in the fluxonium regime, we do not call it fluxonium.

- 2) Rather, we call it a generalized / engineered flux qubit. The flux qubit is a broader concept, referring to any superconducting qubit that consists of a superconducting loop interrupted by Josephson junctions [4] and possibly a capacitive shunt. In fact, one can view a fluxonium as a special case of a generalized N -junction flux qubit [2].

In more detail: Fluxonium consists of a Josephson junction with Josephson energy E_J shunted by a “super-inductance” L [3]. Typically, a large number of array junctions (more than 80) are employed to realize such a large inductance [1-3]. This large inductance enables very small inductive energy $E_L = \left(\frac{\hbar}{2e}\right)^2 / L$, compared to the Josephson energy E_J , such that the potential energy $U(\varphi) = \frac{1}{2}E_L\varphi^2 - E_J\cos(\varphi - \frac{2\pi\Phi_{ext}}{\Phi_0})$ features a steep double-well profile around the sweet spot ($\Phi_{ext} \approx 0.5\Phi_0$). In turn, such a steep double well profile provides a small 0-1 transition dipole moment and large anharmonicity, but limits the 0-1 transition frequency to a few hundred MHz [2].

In contrast, the array of shunting junctions of our qubit are not intended to assume such a steep double-well potential energy profile, but rather a single-well potential at the sweet spot like C-shunt flux qubit [4]. Specifically, we chose the design parameters (e.g., the number of Josephson junctions forming the array, the size of the Josephson junctions, and the shunt capacitance) to engineer a single-well potential that has favorable features such as larger anharmonicity ($\sim 1\text{GHz}$) and higher qubit frequency ($\sim 3\text{GHz}$) enabling fast microwave control.

Lastly, as stated on line 76-78 in our manuscript, we would like to stress that the validation of bispectrum estimation protocol in our work is not limited to the specific type of superconducting qubits. As the Referee kindly pointed out, we engineer non-Gaussian dephasing by exploiting the non-linear dispersion relation of the qubit with respect to the flux, which are common to any flux-tunable superconducting qubits. Therefore, the verification of the bispectrum estimation reported here is not limited to our specific qubit design, but can be reproducible in any type of flux-tunable superconducting qubits including C-shunt flux qubits [4], fluxoniums [1-3], and flux-tunable transmons [6,7].

[1] V. Manucharyan *et al.* Fluxonium: Single Cooper pair circuit free of charge offsets, *Science* **326**, 113-116 (2009).

[2] L. B. Nguyen *et al.* The high-coherence fluxonium qubit, *arXiv*, 1810.11006 (2018).

[3] I. Pop *et al.* Coherent suppression of electromagnetic dissipation due to superconducting quasiparticles, *Nature* **508**, 369-372 (2014).

[4] Clarke, J. & Wilhelm, F. K. Superconducting quantum bits. *Nature* **453**, 1031 (2008).

[5] F. Yan *et al.* The flux qubit revisited to enhance coherence and reproducibility, *Nature Communications* **7**, 12964 (2016).

[6] R. Barends *et al.* Superconducting quantum circuits at the surface code threshold for fault tolerance, *Nature* **508**, 500-503 (2014).

[7] M. D. Hutchings *et al.* Tunable superconducting qubits with flux-independent coherence, *Phys. Rev. Applied* **8**, 044003 (2017).

As a minor point, I believe the bispectrum $S_2(\omega)$ on line 156 needs a vector symbol over the ω .

We thank the Referee for pointing out this inaccuracy. It has been corrected in the revised version.

REVIEWERS' COMMENTS:

Reviewer #1 (Remarks to the Author):

Authors have satisfactorily addressed the issues raised in my report and added relevant references. I recommend the manuscript for publication in the present form in Nature Communication.

Reviewer #2 (Remarks to the Author):

The authors addressed my concerns adequately, where the main affects is that they have justified calling their qubit a C-shunt qubit even though it appears to be more similar to the fluxonium. I also appreciated their response about "real noise" versus input noise since this would also change qualitatively with the qubit type. I recommend for publication of the revised manuscript.